# Comparing the accuracy of two diagnostic methods for detection of light *Schistosoma haematobium* infection in an elimination setting in Wolaita Zone, South Western Ethiopia

**Hussein Mohammed**[1]*, **Toby Landeryou**[2], **Melkie Chernet**[1], **Ewnetu Firdawek Liyew**[1], **Yonas Wulataw**[1], **Birhanu Getachew**[1], **Hailemariam Difabachew**[1], **Anna Phillips**[2], **Rosie Maddren**[2], **Alison Ower**[2], **Kalkidan Mekete**[1], **Habtamu Belay**[1], **Tujuba Endrias**[1], **Ufaysa Anjulo**[3], **Geremew Tasew**[1], **Roy Anderson**[2], **Getachew Tollera**[1], **Ebba Abate**[1]

1 Bacterial, Parasitic and Zoonotic Diseases Research Directorate, Ethiopian Public Health Institute, Addis Ababa, Ethiopia, 2 London Centre for Neglected Tropical Disease Research, London, United Kingdom, 3 Disease Prevention and Health Promotion Core Process, Ministry of Health, Wolaita, Ethiopia

* hussein_ehnri@yahoo.com

## Abstract

Reagent urinalysis dipstick and filtration have been recommended diagnostic methods for the detection of urogenital schistosomiasis. However, the accurate diagnosis of light infections using these methods presents a major challenge. This study evaluates the diagnosis accuracy of light infection with *Schistosoma haematobium* in study participants living in Wolaita Zone, an area targeted for sustainable control of Schistosomiasis, and ultimately interrupt transmission. Urine samples were collected from children and adults in surveys carried out during baseline and longitudinal sentinel site surveys conducted from 2018 to 2020. All urine samples were tested using a reagent urinalysis dipstick test (Haemastix) to detect microhaematuria with reference urine filtration technique as a proxy for S. *haematobium* infection. Sensitivity and specificity were determined in diagnosing urogenital schistosomiasis. Cohen's Kappa statistics was done for the agreement of these diagnostic methods. A total of 12,102 participants were enrolled in the current baseline study. Among them, 285 (2.35%) samples tested positive for microhaematuria and 21 (0.20%) positive for S. *haematobium* eggs. A total of 4,357 samples were examined in year 1 and year 2 using urine dipsticks, and urine filtration 172 (3.95%) and 2 (0.05%) were positive for microhaematuria and S. *haematobium* eggs. The reagent urinalysis dipsticks showed the highest sensitivity and specificity for diagnosing light intensity of infection,100% (95% CI:85.18–100.00) and 97.4% (95% CI: 97.10–97.60), respectively. There is a slight agreement between the two methods (*Kappa* = 0.09, 95% CI: 0.01–0.18). The present study revealed very low prevalence and light intensity of S. *haematobium* infections. The study also highlights that the dipstick test is considered a useful adjunct diagnostic tool for population-based control of urogenital schistosomiasis.

**Data Availability Statement:** All relevant data are within the manuscript and its Supporting Information files.

**Funding:** This research was funded by the Children's Investment Fund Foundation (CIFF), UK through a grant to Ethiopian Public Health Institute, Ethiopia under grant number R-1805-02741. The funders had no role in the design of the study, collection, analysis and interpretation of the data. The finding and conclusion of the study reflect the view of the Authors only.

**Competing interests:** The authors have declared that no competing interests exist.

## Introduction

Human schistosomiasis affects almost 240 million people worldwide, with more than 700 million people at risk in endemic areas. The infection is prevalent in tropical and subtropical areas, particularly within poor communities without potable water and adequate sanitation [1]. In Ethiopia, both intestinal and urinary schistosomiasis are endemic, with the majority of Schistosome infections caused by *S. mansoni* [2, 3]. Limited studies have reported the distribution of urogenital schistosomiasis in Ethiopia. What is available, reports a highly focal distribution, predominantly in both the northeastern regions, following the middle and upper Awash River Valley [4, 5], and in eastern regions proximal to the lower Wabe-Shebele valleys [6, 7]. There are also reports of infection in western regions such as Kurmuk on the Ethio-Sudan border [8, 9]. The spread of the disease has been linked to population movement, and fluctuations in water contact behaviour resultant from both natural geological events, and anthropogenic interventions [10].

Infection of *S. haematobium* affects both the urinary and genital tracts, with chronic infection resulting in fibrosis of the bladder and ureter that in severe cases can cause bladder cancer and renal failure [11]. *S. haematobium*-associated microhaematuria results from the deposition of the parasite's spiny eggs in the submucosa, causing bladder lesions, the detection of microhaematuria is performed using a reagent urinalysis dipstick [12, 13].

The most common methods for diagnosing urogenital schistosomiasis are reagent urinalysis dipstick, urine filtration, and point-of-care urine circulating cathodic antigen (POC-CCA) assay examination of urine samples. The reagent urinalysis dipstick has long been recommended as a relatively inexpensive and accurate proxy for the detection of *S. haematobium* infection [14]. Following detection of microhaematuria using the dipstick, the diagnosis of *S. haematobium* is confirmed through urine filtration and microscopy to identify and quantify the presence of *S. haematobium* eggs in urine [15]. Several studies confirmed reagent urinalysis dipstick against standard urine filtration and detection of microhaematuria is an adjunct for estimation of urogenital schistosomiasis and related morbidity [16, 17].

The current Federal Ministry of Health (FMoH) recommendations for schistosomiasis control is based upon mass praziquantel administration of endemic woredas (districts), targeting school-aged children (SAC) or high-risk communities [18]. Parasitological mapping conducted by the Ethiopian Public Health Institute (EPHI) informs the FMoH with the endemic woredas requiring treatment [19]. This mapping is dependent on the use of reagent urinalysis dipstick and urine filtration to determine each woreda's endemicity status. For the current national program use it would be essential to know the sensitivity and accuracy of the test, determining the epidemiological status of woredas, especially in low prevalence settings [19, 20].

The low sensitivity and specificity of diagnostics for other NTDs, such as soil-transmitted helminths, in low prevalence settings has been noted in previous publications [21], presenting a major hindrance in accurately recording low or no parasite prevalence. However, there is currently a paucity of information regarding the efficacy of reagent urinalysis dipstick as a diagnostic tool in low diagnostic settings for schistosomiasis. The available data of its diagnostic performance is reported with varying sensitivity and accuracy by different research groups [18, 20]. These potential limitations associated with reagent urinalysis dipstick in low prevalence regions [22], may impose a challenge for accurate diagnosis of microhaematuria. This morbidity associated with light infections should be taken seriously, as the host's immune response to eggs, not the number of worm pairs, causes schistosomiasis pathology [23].

In Ethiopia, few studies have been conducted to compare the performance of reagent urinalysis dipstick against urine filtration in different transmission settings [5, 24]. This study will seek to evaluate the sensitivity and specificity of reagent urinalysis strips, urine filtration, and

POC-CCA and thus their suitability in large-scale mapping and evaluation activities for detecting microhaematuria for *S. haematobium* infections. This work has been conducted as part of the work of the Geshiyaro project, described previously by Mekete et al [25].

## Materials and methods

### Ethics statement

This study was conducted as part of the Geshiyaro project, which was ethically approved by Institutional Review Board (IRB) of the Ethiopian Public Health Institute (EPHI). Permission letters have been secured from Regional, Zonal Health Offices, and Educational bureaus. They have been shared with the district Health Office and community administrators and school directors. Informed consent was obtained from study participants after detailed information was provided in the local language on the aims and procedures of the study, and documented electronically through a data collection form. Verbal consent was obtained from parents or guardians on behalf of any enrolled children.

### Study sites

The study was conducted within the Wolaita Zone, in the Southern Nations Nationalities People Regional (SNNPR) state of Ethiopia, located 300 kilometers south of Addis Ababa., SNNPR consists of 22 districts or 'woredas' (the smallest functional unit of the Ethiopian health care system, covering a population of at least 100,000), and has a population of approximately 1.9 million. The Wolaita Zone is boarded to the southwest by the Omo River, and in the east by the Bilate river and Lake Abaya. Urogenital schistosomiasis data is not available in Wolaita Zone.

### Sample size determination

**Sample size calculations.** The sample size was estimated based on the probability that at least 70% of woredas had a true schistosomiasis prevalence of 15% (classified as 10% or over), under the assumption that the lower focality of schistosomiasis often leads to underestimated prevalence due to insufficient sampling. Due to varying number of kebeles per woreda, between 26% and 75% of kebeles were surveyed within each woreda [23], with a higher proportion of kebeles being surveyed in small woreda.

**Number of kebeles and number of individuals.** A cross-sectional baseline mapping survey took place, consisting of 13,000 individuals. One hundred and thirty-seven kebeles from selected woredas were randomly identified. Within each selected kebele, a parasitological mapping survey was conducted, sampling 100 individuals per community, stratified for equal sample size (20 individuals) by age grouping (preschool-age children (pre-SAC) (1–4 years), SAC (5–14 years), 15–20 years, 21–35 years, 36+ years) and sex, were randomly selected in each kebele. In addition, a longitudinal sentinel site cohort of 4,500 individuals, were followed across 30 kebeles, whereby 150 individuals were selected in age and sex stratified groupings as presented above, and parasitological data were monitored on an annual basis.

### Field procedures

Data detailing participant demographics and WaSH infrastructure access were collected from consenting participants during the baseline and sentinel site surveys. Biological samples (including 10ml urine) were collected and logged against individual barcoding system to link participant meta-data with parasitological mapping data. The collected samples were transported to the nearby health center for laboratory analysis. Electronic data from each survey

was collected in the field using Samsung J5 smartphones. Questionnaires were coded and uploaded using the SurveyCTO sever. The microhaematuria, urine filtration and POC-CCA test results were recorded on paper laboratory forms, and later transcribed electronically to SurveyCTO at the end of each day.

## Detection of urine for microhematuria

Reagent urinalysis dipsticks (Haemastix®, Siemens, Healthcare Diagnostics, United Kingdom) were used following the manufacturer's instructions. Principally, each dipstick was dipped into the barcode urine container for 5 seconds. After 1 minute, the resulting colour change of the strip was compared with manufacturer's diagnostic colour chart. Intensity of colour change is proportional to the amount of blood in the urine, and results were scored into 6 diagnostic categories; negative, trace non-haemolysed, trace haemolysed, +, ++, +++ according to the manufacturer recommendations. Trace-positive reagent urinalysis strips were considered positive (S1 Appendix).

## Urine filtration technique

Two microscope filtration slides were prepared from a single urine sample. Each urine sample was rigorously shaken to ensure mobilization of any eggs present, then 10 ml of urine was drawn into a syringe and passed out against a nylon filter with a pore size of 20 μm, twice. The two prepared urine filters were placed on microscope slides, using a drop of Lugol's iodine to stain *S.haematobium* eggs for identification under the microscope [26]. Each slide was read by different laboratory technician. The presence and quantity of *S. haematobium* eggs was recorded for each microscope slide, and finally reported as an average of two slides as egg count/10 ml urine. A participant was considered = negative if no *S. haematobium* eggs were detected on either slide Any discordant results between laboratory technicians were re-examined by a senior laboratory technician.

## POC-CCA cassette test

The POC-CCA cassette test was blindly performed on all urine samples according to the manufacturer's instructions (Rapid Medical Diagnostics, South Africa). One drop of urine was added to the sample well, and test results were read 20 minutes afterwards. In instances where control bands did not develop, the cassette test was considered invalid, a second cassette test was undertaken. The valid POC-CCA tests were scored as negative (only control line), trace (trace test line), +1 (test line weakly visible), +2 (test line same opacity as the control line) and +3 (test line darker than control line). Trace POC-CCA test results were considered as positive.

## Data analysis

*S. haematobium* infection intensity was assessed using the WHO classification, and classed into low infestation (1–49 eggs/10 ml of urine) and high infestation ≥50 eggs/10 ml urine [26]. Data were analysed using R version 4.0.4 (R Foundation for Statistical Computing, Vienna, Austria).

Diagnostic performance of urine reagent urinalysis dipstick was determined by calculating sensitivity, specificity, positive and negative predictive values, and accuracy. Using the urine filtration method as the gold standard, sensitivity of urine reagent urinalysis dipstick was calculated as the proportion of positives that were correctly identified when compared with the standard test. The specificity of the test was calculated as the proportion of negatives that were correctly identified when compared with the diagnostic reference test. Positive predictive

value (PPV) describes the probability that a disease is present when the test is positive. The negative predictive value (NPV) describes the probability that the disease is not present when the test is negative. The agreement between the urine filtration and reagent urinalysis dipstick was determined using *Cohen's kappa* (κ)-statistics. The accuracy was the overall probability that a study participant was correctly classified by both diagnostic tests. *P*-value <0.05 was considered statistically significant.

## Results

As shown in Fig 1 below, a total of 16,780 children and adults were enrolled to participate in the baseline and cross-sectional surveys conducted between 2018 and 2020. During baseline mapping survey, 12,102 samples were examined using dipsticks. Among them, 285 (2.35%) samples were positive for microhaematuria and 21 (0.17%) positive for *S. haematobium* eggs by urine filtration.

A total of 501 urine samples were examined in 2019 sentinel sites Year 1 survey. 10 samples (2.00%) were positive for microhematuria, and all of them were negative for *S. haematobium* infection. A total of 3,875 individuals participate in the sentinel sites Year 2 survey conducted in 2020, and 3,856 underwent for dipstick test. 162 samples (4.20%) were positive for microhaematuria, and 2 (0.05%) were positive for *S. haematobium*. Overall, 23 individuals were positive (0.14%) by urine filtration diagnostics. Of these, 21 (0.13%) individuals were positive by POC-CCA cassette test.

The prevalence of microhaematuria infection based on reagent urinalysis dipstick was comparable between males (49.1%) and females (50.9%), and across the different age groups pre-SAC (17.1%), SAC (25.0%), 15 to 20 (22.2%), 21 to 35 (18.9%) and 35+(16.9%) years. The prevalence of microhaematuria was higher in SAC (4–15 years age) compared to other age groups as shown in Fig 2.

The sensitivity and specificity of reagent urinalysis dipstick was 100% (95% CI:85.2%–100.00%) and 97.4% (95% CI: 97.1%-97.6%), respectively. The PPV was low (5.03%), and NPV was high (100%) as shown in Table 1. All infections were classed as light infection by urine filtration (1–49 egg/10 ml urine). A false-positive reagent urinalysis strip test result was noted in 457(2.8%) of the study participants.

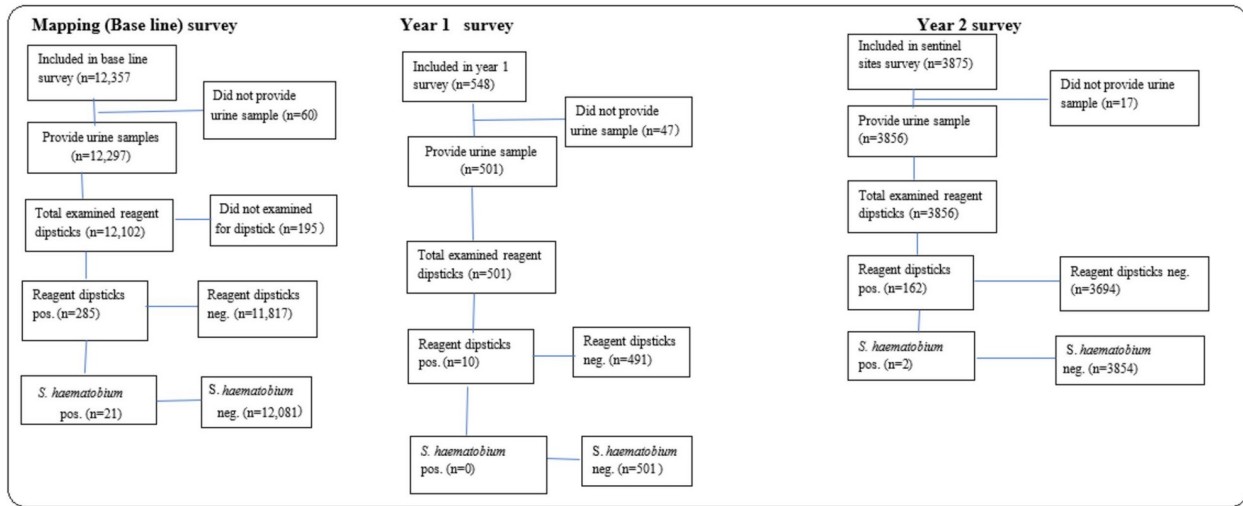

**Fig 1. Flowchart of study phase participation, sample provision and diagnostic tests performed during the study period.**

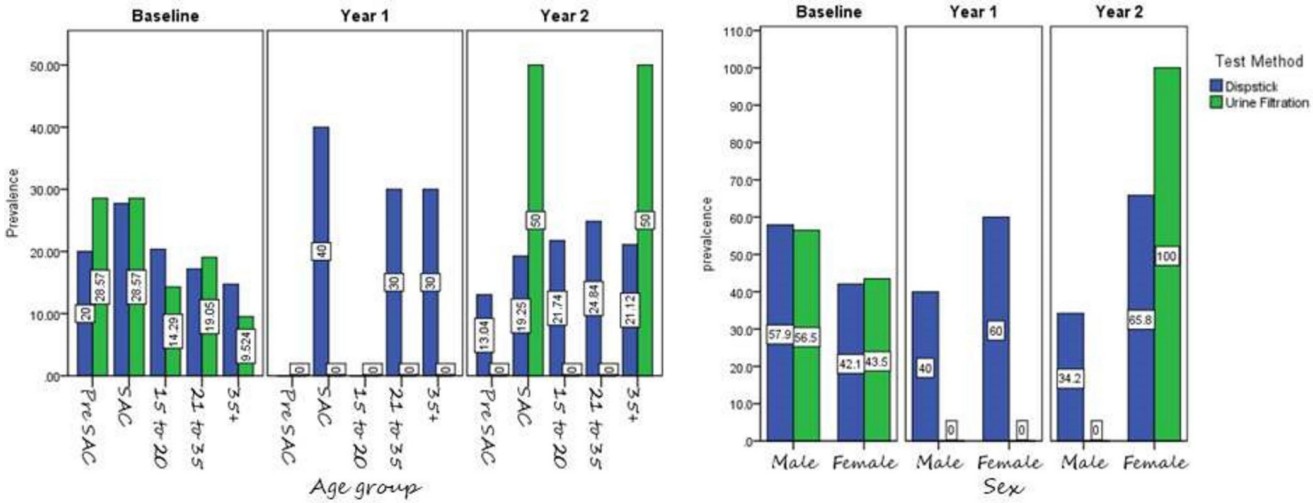

**Fig 2. Prevalence of microhaematuria and *S haematobium* by study phase age category (left panel and sex (right panel).**

The reagent urinalysis dipstick testing showed a better diagnostic at trace (trace non-haemolysed and haemolysed) compared to other microhaematuria range, as shown in Table 2. The agreement between the two methods in diagnosing for the presence of microhaematuria and *S. haematobium* egg was slight (*Kappa* = 0.09 (95%CI: 0.01–0.18). That is, the likelihood of detecting a negative/positive result is unlikely.

## Discussion

This study was conducted to determine the diagnostic accuracy of reagent urinalysis dipstick for microhaematuria in a *S. haematobium* in low prevalence setting. The quick nature of the laboratory diagnostic, paired with its low cost, makes the urinalysis reagent dipstick suitable for estimating *S. haematobium* prevalence in a community-level diagnostic screening surveys for schistosomiasis control programs.

The Cohen's Kappa statistic between the two diagnostics, reagent urinalysis dipstick and urine filtration, revealed a slight agreement (*kappa* = 0.09). Robinson *et al* inferred that the reagent urinalysis dipstick test showed higher sensitivity than that of urine filtration in assumed low *S. haematobium* prevalence settings [27]. This study similarly concludes that the sensitivity and specificity of the reagent urinalysis dipstick is a good diagnostic tool for low *S.*

**Table 1. Sensitivity, specificity and predictive values of the reagent dipstick method for *S haematobium* in study participants when filtration results are considered reference test.**

| Urine filtration | | Microhematuria | | |
|---|---|---|---|---|
| | Positive | Positive | Negative | Total |
| | | 23 | 0 | 23 |
| | Negative | 434 | 16002 | 16436 |
| | Total | 457 | 16002 | 16459 |
| Sensitivity of dipstick | | 100% | | |
| Specificity of dipstick | | 97.4% | | |
| Positive predictive value (PPV) | | 5.03% | | |
| Negative predictive value (NPV) | | 100% | | |
| Accuracy | | 97.4% | | |

**Table 2. Association of urine filtration and microhematuria.**

| Microhaematuria positive | n(%) | Urine filtration, n(%) | p-value |
|---|---|---|---|
| trace non-haemolysed | 122(31.7) | 1(4.3) | 0.997 |
| trace haemolysed | 48(12.5%) | 4(17.4) | 0.997 |
| + | 109(28.3) | 10(43.5) | 0.997 |
| ++ | 52(13.5) | 5(21.7) | 0.997 |
| +++ | 54(14.0) | | 0.996 |
| Total | 385 | 23 | |

*haematobium* prevalence settings [28]. Furthermore, the study demonstrated that trace positive microhaematuria (considered as trace non-haemolysed and haemolysed results) provided good diagnostic sensitivity, corroborating the results of a previous study also conducted in Ethiopia [25].

According to WHO classification, this study also revealed that the prevalence and infection intensity categorized as light infection (cut-off point < 50 eggs per 10 mL), establishing infection intensity can help in guiding the future treatment strategies [26]. In this study, some microhaematuria results could not be associated with *S. haematobium* infection through reference egg filtration. This is most likely due to either low prevalence setting or contamination of urine by menses, genital mutilation, menstrual blood or pregnancy in females; these are previously established confounding factors that can affect urine reagent urinalysis dipstick results [16, 29–31]. The combination of confounding factors indicate that reagent urinalysis dipstick cannot be used as a single diagnostic tool when confirming infection of *S. haematobium* and a continued combination of urine filtration is needed.

This study also shows that the PPV was low (5.03%) in low prevalence settings indicating that the majority microhaematuria positive individuals in this locality are not related to urogenital schistosomiasis. The predictive values are generally reliant on the prevalence of endemicity [32], lower PPV and the higher NPV indicating the lower prevalence of the disease in given locations [33, 34]. Similarly, our findings also showed low PPV and high NPV. These might indicate an association between the predictive values and endemicity.

Urine filtration alone revealed prevalence of S. haematobium in 23 (0.14%,) individuals and reagent urinalysis dipstick testing for microhaematuria was found in 172 (1.03%). However, a study conducted in western Ethiopia [8], with high transmission area, urine filtration and microhaematuria were reported 57.8% and 68.6%, respectively. The geographic variance of S. haematobium prevalence in Ethiopia again indicates the highly focal infective presence of Schistosoma, throughout endemic countries [18].

Children had greater prevalence microhaematuria than adults with the same infection intensity [29, 30]. Similarly, we found microhaematuria was highest in school age group (4–15 years age) than adults. This may suggest that adult experience release less blood in response to parasite egg deposition around urinary tract [35].

The POC-CCA assay has also shown effective in diagnosing urinary schistosomiasis [36]. However, studies on the use of POC-CCA for *S. haematobium* present conflicting results. Previous study found POC-CCA less sensitive and specific than the urine filtration and dipstick methods in the diagnosis of *S. haematobium* infection in endemic areas [37]. In contrast a study by Said et al. [38] observed a significantly higher sensitivity in the diagnosis of both *S. haematobium* and *S. mansoni* by this method as compared to egg counts in urine and stool, respectively. Similarly, the current study found twenty-three positive *S. haematobium* eggs observed by urine filtration test, which was also detected in 21 (0.13%) positives with the POC-CCA test. This suggests that the result of POC-CCA tests were considered similar to the

results of urine filtration. The definitive use of a single test to diagnose *S. haematobium* is still waiting verification. POC-CCA is a frequently used antigen detection test, however, numerous studies have revealed a mixed outlook into the replacement of microscopy with antigen test alone [39, 40]. The field-based diagnosis of *S. haematobium* indicates that a combined approach is still necessary, using antigen, dipstick and microscopy to achieve the highest degree of sensitivity to confirm infection. The egg output of S. *haematobium* shows a considerable day-to-day fluctuation of released egg in urine could vary within days [41]. Thus, the follow-up diagnosis or repeated urine sampling could improve sensitivity and increase the probability of detecting light infections.

This study relies only on a single day urine sample, hence repeated daily sampling of urine may be of consideration when surveying low prevalence settings to confirm potential transmission break of the disease [42].

## Conclusion

The results of this study indicates that the reagent urinalysis dipstick is considered a useful adjunct for community-based control in urogenital schistosomiasis areas. The preliminary confirmation of microhaematuria followed by urine filtration can aide in the efficiency of parasitological surveys in region-wide baseline and sentinel site surveys. However, the undetected *S. haematobium* may be due to the lack of sensitivity of widely used urine filtration tool. To further increase the evaluation of the diagnostic performance of reagent urinalysis dipsticks, its comparison against molecular diagnostic workflows would help in particularly in regions with targeted transmission break interventions [43], evaluation the accuracy performance of reagent urinalysis dipstick method, is therefore crucial for rapid diagnosis of urogenital schistosomiasis in areas of light transmission settings.

## Supporting information

**S1 Appendix. Results of number of individuals detected by POC-CCA test in Wolaita Zone, Ethiopia.**
(RTF)

**S2 Appendix. Minimum data set.**
(DOCX)

## Acknowledgments

The authors would like to thank the study participants and data collectors, supervisors, funders, zone and woreda health offices and local guides.

## Author Contributions

**Conceptualization:** Hussein Mohammed, Alison Ower, Kalkidan Mekete, Tujuba Endrias, Ufaysa Anjulo, Ebba Abate.

**Data curation:** Hussein Mohammed, Birhanu Getachew, Hailemariam Difabachew, Rosie Maddren, Habtamu Belay, Geremew Tasew.

**Formal analysis:** Melkie Chernet, Ewnetu Firdawek Liyew.

**Investigation:** Yonas Wulataw.

**Supervision:** Hussein Mohammed, Roy Anderson, Getachew Tollera, Ebba Abate.

**Validation:** Hussein Mohammed.

**Visualization:** Hussein Mohammed.

**Writing – original draft:** Hussein Mohammed, Toby Landeryou, Roy Anderson, Ebba Abate.

**Writing – review & editing:** Hussein Mohammed, Anna Phillips.

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
