## [Decision Letter · Decision Letter 0]

6 May 2021

PONE-D-21-07166

Comparing the diagnostic accuracy for detection of light Schistosoma haematobium infection in an elimination setting in Wolaita Zone, South Western Ethiopia.

PLOS ONE

Dear Dr. Mohammed,

Thank you for submitting your manuscript to PLOS ONE. After careful consideration, we feel that as submitted it does not fully meet PLOS ONE’s publication criteria in its present form. The subject topic is of significance and we invite you to submit a revised version of the manuscript that addresses all the points raised during the review process. Please note, your revised manuscript will undergo further review before reaching a final decision.

Please submit your revised manuscript within 60 days from the date of this letter. If you will need more time than this to complete your revisions, please reply to this message or contact the journal office at plosone@plos.org. Please include the following items when submitting your revised manuscript:

We look forward to carefully revised and edited manuscript for further consideration.

Kind regards,

Nirbhay Kumar, Ph. D.

Academic Editor

PLOS ONE

Journal Requirements:

3. Please include your tables as part of your main manuscript and remove the individual files. Please note that supplementary tables (should remain/ be uploaded) as separate "supporting information" files.

4. We suggest you thoroughly copyedit your manuscript for language usage, spelling, and grammar. If you do not know anyone who can help you do this, you may wish to consider employing a professional scientific editing service.  

5. Please provide additional details regarding participant consent. In the ethics statement in the Methods and online submission information, please ensure that you have specified how verbal consent was documented and witnessed.

7. Please amend your list of authors on the manuscript to ensure that each author is linked to an affiliation. Authors’ affiliations should reflect the institution where the work was done (if authors moved subsequently, you can also list the new affiliation stating “current affiliation:….” as necessary).

8. We note you have included a table to which you do not refer in the text of your manuscript. Please ensure that you refer to Table 1 in your text; if accepted, production will need this reference to link the reader to the Table.

9. Thank you for stating the following in the Acknowledgments Section of your manuscript:

[The authors would like to thank the study participants and data collectors, supervisors, **funders**, zone and weredahealth offices and local guides.]

 [The authors received no specific funding for this work.]

Reviewers' comments:

Reviewer's Responses to Questions

**Comments to the Author**

1. Is the manuscript technically sound, and do the data support the conclusions?

Reviewer #1: Partly

Reviewer #2: Yes

2. Has the statistical analysis been performed appropriately and rigorously? 

Reviewer #1: I Don't Know

Reviewer #2: No

3. Have the authors made all data underlying the findings in their manuscript fully available?

Reviewer #1: Yes

Reviewer #2: Yes

4. Is the manuscript presented in an intelligible fashion and written in standard English?

Reviewer #1: Yes

Reviewer #2: Yes

5. Review Comments to the Author

Reviewer #1: This study is a relevant comparison of the diagnostic accuracy of detecting light Schistosoma haematobium infections which would go a long way towards addressing shortcomings in the monitoring and evaluation of intervention programs. The study compares the urinalysis dipstick method and the urine filtration method (reference gold-standard) statistically using a significantly large study population over a period of two years. The study was well designed to compare the two methods which revealed significantly different outcomes in terms of sensitivity and specificity. The study addresses a very relevant question of the accuracy of dipstick method and the possibility of its use as a proxy or adjunct diagnostic tool for the more involving and less sensitive gold-standard filtration method. (Continue to attachment)

Reviewer #2: This manuscript is a study on two diagnostic techniques for the detection of low prevalence of Schistosoma haematobium infection. The reviewer's comments have provided and they attached here with this submission

6. PLOS authors have the option to publish the peer review history of their article (what does this mean?). If published, this will include your full peer review and any attached files.

Reviewer #1: No

Reviewer #2: No

---

## [Author Response · Author response to Decision Letter 0]

9 Aug 2021

We respond the reviewers comments point by point

---

## [Decision Letter · Decision Letter 1]

7 Feb 2022

PONE-D-21-07166R1Comparing the diagnostic accuracy for detection of light Schistosoma haematobium infection in an elimination setting in Wolaita Zone, South Western Ethiopia.

PLOS ONE

Dear Dr. Hussein Mohammed,

Thank you for submitting your manuscript to PLOS ONE. Your manuscript has been reviewed by number required reviewers.  The reviewers have brought up some minor concerns  in material and methods, results and conclusion in the text manuscript. Please carefully consider that reviewer(s) comment(s) and address them point-by-point before revision submission.

We look forward to receiving your revised manuscript.

Kind regards,

Wannaporn Ittiprasert, Ph.D

Academic Editor

PLOS ONE

Journal Requirements:

Reviewers' comments:

Reviewer's Responses to Questions

**Comments to the Author**

1. If the authors have adequately addressed your comments raised in a previous round of review and you feel that this manuscript is now acceptable for publication, you may indicate that here to bypass the “Comments to the Author” section, enter your conflict of interest statement in the “Confidential to Editor” section, and submit your "Accept" recommendation.

Reviewer #1: All comments have been addressed

Reviewer #2: All comments have been addressed

2. Is the manuscript technically sound, and do the data support the conclusions?

Reviewer #1: Partly

Reviewer #2: Partly

3. Has the statistical analysis been performed appropriately and rigorously? 

Reviewer #1: I Don't Know

Reviewer #2: Yes

4. Have the authors made all data underlying the findings in their manuscript fully available?

Reviewer #1: Yes

Reviewer #2: Yes

5. Is the manuscript presented in an intelligible fashion and written in standard English?

Reviewer #1: Yes

Reviewer #2: Yes

6. Review Comments to the Author

Reviewer #1: The authors partly addressed my concerns from thr initial review on precisely how they carried out the gold standard urine filtration method. They did not collect 2 or 3 consecutive daily urine samples. Instead, they carried out urine filtration with replicates of the same one-day urine sample. Due to diurnal variation in egg shedding, this does not present a fair assessment of the gold standard urine filtration method with the dipstick microhematuria test. The author's still did not show a table of the POA-CCA test mentioned.

Reviewer #2: Review of manuscript: Comparing the diagnostic accuracy for detection of light Schistosoma haematobium infection in an elimination setting in Wolaita Zone, South Western Ethiopia

Reviewer’s Comments to Author

Comments to the Author

The manuscript proposed by Mohammed et al. is titled: Comparing the diagnostic accuracy for detection of light Schistosoma haematobium infection in an elimination setting in Wolaita Zone, South Western Ethiopia

The study described in this manuscript was conducted in the Wolaita Zone, South Western Ethiopia and it provides information on the comparative analyses of two schistosomiasis diagnostic techniques – Urinalysis dipstick and urine filtration. The authors attempted to shed a new light on the use of these diagnostic techniques in order to fill gaps in our knowledge of schistosomiasis control. The study was conducted between 2018 and 2020 and a total of 16,780 children and adults participated in the study. A total of 12,102 samples were examined in the baseline study using dipstick and urine filtration; among them, 285 (2.4%) samples were positive for microhematuria and 21 (0.2%) positive for S. haematobium eggs by urine filtration. 501 urine samples were examined in 2019 and 10 (2.0%) were positive for microhematuria, and all of them were negative for S. haematobium infection. A total of 3875 individuals were examined in 2020 and 162 (4.2%) were positive for microhematuria and 2 (0.05%) were positive for S. haematobium eggs. The prevalence of microhematuria infection based on reagent dipstick was comparable between males (50.9%) and females (49.1%), and across the different age groups Pre-SAC (17.1%), SAC (25.0%), 15 to 20 (20.4%), 21 to 35 (20.2%) and 36+(17.3%) years. The prevalence of microhematuria was highest in SAC (4-15 years age) compared to other age groups. The sensitivity and specificity of reagent strip was 100% (95% CI:85.18%–100.00%) and 97.4% (95% CI: 97.10%-97.60%), respectively. The PPV was low 5.03% and NPV was high 100%.

Most of the concerns addressed in my initial review have been addressed.

Abstract

The calculations are still not correct. 285/16,780 is not equal to 2.4% (285/12,102 = 2.4%). Also, 172/16,654 is not equal to 6.2%. The calculations need to be checked again.

Materials and Methods

Sample size calculations

“The sample size was estimated using at least 70% probability of woredas with true prevalence of schistosomiasis of 15% being classified as 10% or over, under the assumption that the lower

focality of STH was led to sufficient sample sizes for schistosomiasis between 26% and 75% of

kebeles were surveyed within each woreda [21], with a higher proportion of kebeles being

surveyed in small woreda.”

The sentence above has still not been revised for better comprehension.

Results

There is a great disparity between the samples analyzed during the baseline study, in year 1 and year 2 respectively? Why is this so? Why are there only 501 samples analyzed in year 1?

The information in table 1 should be represented graphically

Discussion

The discussion section still needs a careful revision to correct typographical errors. The sentence “Using kappa value evaluation between the two tests was revealed slight agreement” should be corrected.

Conclusion

It is still difficult to understand the conclusion from the study based on the wording of the sentences. The authors need to compose the information in a way that is clear and understandable.

The manuscript as revised is an improvement on the previous version but it still requires further revision.

7. PLOS authors have the option to publish the peer review history of their article (what does this mean?). If published, this will include your full peer review and any attached files.

Reviewer #1: **Yes: **Davison Sangweme

Reviewer #2: No

---

## [Author Response · Author response to Decision Letter 1]

23 Mar 2022

we are response the reviewer and editor comments

---

## [Decision Letter · Decision Letter 2]

8 Apr 2022

Comparing the accuracy of two diagnostic methods for detection of light Schistosoma haematobium infection in an elimination setting in Wolaita Zone, South Western Ethiopia

PONE-D-21-07166R2

Dear Dr. Hussein Mohammed,

We’re pleased to inform you that your manuscript has been judged scientifically suitable for publication and will be formally accepted for publication once it meets all outstanding technical requirements.

Kind regards,

Wannaporn Ittiprasert, Ph.D

Academic Editor

PLOS ONE

Additional Editor Comments (optional):

Reviewers' comments:

Reviewer's Responses to Questions

**Comments to the Author**

1. If the authors have adequately addressed your comments raised in a previous round of review and you feel that this manuscript is now acceptable for publication, you may indicate that here to bypass the “Comments to the Author” section, enter your conflict of interest statement in the “Confidential to Editor” section, and submit your "Accept" recommendation.

Reviewer #2: All comments have been addressed

2. Is the manuscript technically sound, and do the data support the conclusions?

Reviewer #2: Yes

3. Has the statistical analysis been performed appropriately and rigorously? 

Reviewer #2: Yes

4. Have the authors made all data underlying the findings in their manuscript fully available?

Reviewer #2: Yes

5. Is the manuscript presented in an intelligible fashion and written in standard English?

Reviewer #2: Yes

6. Review Comments to the Author

Reviewer #2: The authors have addressed the concerns in my previous review. Most of the typographical errors have been corrected and the statistical analyses have been improved upon. The manuscript is a major improvement on the previous versions

7. PLOS authors have the option to publish the peer review history of their article (what does this mean?). If published, this will include your full peer review and any attached files.

Reviewer #2: **Yes: **Mobolaji Okulate

---

## [Editor Report · Acceptance letter]

21 Apr 2022

PONE-D-21-07166R2 

Comparing the accuracy of two diagnostic methods for detection of light *Schistosoma haematobium* infection in an elimination setting in Wolaita Zone, South Western Ethiopia 

Dear Dr. Mohammed:

I'm pleased to inform you that your manuscript has been deemed suitable for publication in PLOS ONE. Congratulations! Your manuscript is now with our production department. 

Kind regards, 

on behalf of

Dr. Wannaporn Ittiprasert 

Academic Editor

PLOS ONE